# Psychedelic 5-HT2A agonist increases spontaneous and evoked 5-Hz oscillations in visual and retrosplenial cortex
Callum M. White[1,2,5], Zohre Azimi[1,5], Robert Staadt[1,5], Chenchen Song [3], Thomas Knöpfel [4,6] ✉ & Dirk Jancke [1,2,6] ✉

Visual perception appears largely stable in time. However, psychophysical studies have revealed that low frequency (0.5 – 7 Hz) oscillatory dynamics can modulate perception and have been linked to various cognitive states and functions. Neither the contribution of waves around 5 Hz (theta or alpha-like) to cortical activity nor their impact during aberrant brain states have been resolved at high spatiotemporal scales. Here, using cortex-wide population voltage imaging in awake mice, we found that bouts of 5-Hz oscillations in the visual cortex are accompanied by similar oscillations in the retrosplenial cortex, occurring both spontaneously and evoked by visual stimulation. Injection of psychotropic 5-HT2AR agonist induced a significant increase in spontaneous 5-Hz oscillations, and also increased the power, occurrence probability and temporal persistence of visually evoked 5-Hz oscillations. This modulation of 5-Hz oscillations in both cortical areas indicates a strengthening of top-down control of perception, supporting an underlying mechanism of perceptual filling and visual hallucinations.

Oscillatory activity of cortical neuronal populations at various frequencies have been shown to be associated with brain state-dependent cognition and perception[1–4]. The involvement of low frequency oscillations in visual processing has been particularly well studied. Cortical theta (4–7 Hz) and alpha-like (3–5 Hz) oscillations are hypothesized to reflect rhythmic sampling[5], modulations of attention[6] and memory consolidation during periodic replay[7–9]. In mice, alpha-like oscillations strongly correlate with activity in the visual thalamus[9]. More recently, it has been suggested that familiarity of visual stimuli can frequently trigger and modulate theta oscillations in the primary visual cortex (V1) of mice[10–14].

Neuromodulators regulate the dynamics of cortical interactions[15] and thereby shape the brain-wide rhythms of activity[16–20]. For instance, serotonin (5-HT) suppresses theta oscillations in the hippocampus[21,22] and modulates the strength of evoked activity in the visual cortex[23–27], while hippocampal oscillations have also been shown to actively modulate neuronal activity within V1[28]. Elevated 5-HT signaling can induce deviations from normal visual perception[18]. A well-known case is the acute effects of serotonergic psychedelics, which can lead to profound changes in perception, cognition, mood[29,30], and motor-related behaviors such as hyperactivity, stereotypy[24], and head-twitch response in mice[31–33]. In humans,

psychedelic actions are often accompanied by visual hallucinations. Visual hallucinations can also appear as a symptom of certain brain disorders[34], such as during episodes of psychosis, where visual hallucinations are accompanied by reduced activity in V1 (external inputs) and perception driven more strongly by internal mechanisms[35]. However, how serotonergic psychedelics act on cortical circuit dynamics is largely unknown.

Currently, 5-HT-dependent changes in rhythmic brain activities[36–41] have mainly been derived from observations using electrical measurements of brain activity (e.g. using electroencephalography or local field potential recordings). However, such methods are limited by spatial resolution and/or coverage and capture activity blind to cell type. To overcome these limitations, here we used optical imaging of genetically encoded voltage indicators (GEVIs) in pyramidal neurons, which enables direct monitoring of cortex-wide voltage activity selectively from pyramidal cells in awake head-fixed mice[42,43], using a cortex-wide preparation and a mesoscopic imaging approach that achieves high spatial coverage. In contrast to calcium imaging approaches, mesoscopic voltage imaging readily resolves subthreshold, suprathreshold and hyperpolarizing population activities.

Capitalizing on this powerful approach, we first observed episodes of 5-Hz oscillatory activity occurring in V1 both spontaneously and evoked by

[1]Optical Imaging Group, Institut für Neuroinformatik, Ruhr University Bochum, Bochum, Germany. [2]Monoaminergic Neuronal Networks & Diseases (MoNN&Di), Ruhr University Bochum, Bochum, Germany. [3]Lee Kong Chian School of Medicine, Nanyang Technological University, Singapore, Singapore. [4]JC STEM Laboratory for Neuronal Circuit Dynamics, Hong Kong Baptist University, Hong Kong (SAR), China. [5]These authors contributed equally: Callum M. White, Zohre Azimi, Robert Staadt. [6]These authors jointly supervised this work Thomas Knöpfel, Dirk Jancke. ✉e-mail: tknopfel@knopfel-lab.net; dirk.jancke@rub.de

visual simulation. Through the high spatial coverage of our approach, we detected similar and co-occurring oscillatory activity in the retrosplenial cortex (RSC), indicating a coupling between these two regions. Then, we administered a serotonergic psychedelic to investigate how 5-HT receptor activation modulates the 5-Hz oscillatory activity in the cortex. We observed that properties of spontaneous and sensory evoked oscillatory episodes are facilitated by an injection of a 5-HT2AR agonist (either 2,5-Dimethoxy-4-iodoamphetamine (DOI) or TCB2). Collectively, the present study advances our concepts of how serotonergic psychedelics alter perception and consciousness.

## Results

### 5-Hz oscillations occur in primary visual cortex (V1) pyramidal neurons both spontaneously and evoked by a visual stimulus

We performed widefield optical voltage imaging in awake mice. To facilitate stable optical recordings, the mice were head-fixed but otherwise allowed to run freely on a treadmill (Fig. 1a). All mice were first habituated to the setup

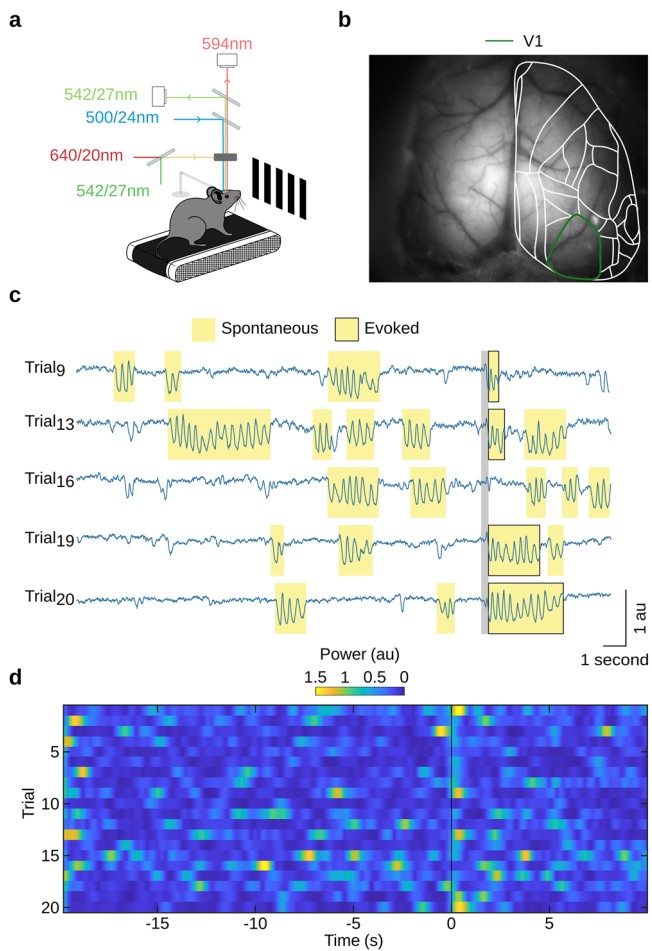

**Fig. 1 | Spontaneous and stimulus-evoked 5-Hz oscillations are observed in V1 voltage fluctuations. a** Experimental configuration for FRET optical imaging in head-fixed awake mice (see "Methods" section). **b** Widefield image of cortical vasculature in a transcranial cortex-wide window preparation, with cortical regions of the right hemisphere (white outline) estimated using the Allen Mouse Brain Atlas[77] (Supplementary Fig. 1). Green outline indicates primary visual cortex (V1). **c** Examples of single trial V1 population voltage activity (spatially averaged across V1 marked green in **b**). Episodes of 5-Hz oscillations occurring both spontaneously (yellow highlight) and evoked (yellow highlight with black outline) by a visual stimulus (moving grating at 100% contrast, grey vertical bar marks onset and duration). **d** Time-resolved 5-Hz power in V1 population voltage activity in individual trials across a single imaging session (same session as that shown in **c**). Visual stimulus presented at time zero (vertical line) for a duration of 0.2 s.

prior to optical imaging but were not exposed to any specific stimuli or task training.

We measured ongoing and evoked changes in membrane potential across both cortical hemispheres using FRET optical imaging (Fig. 1a, b) of layer II/III or II/III/V pyramidal neurons that express genetically encoded voltage indicators[42,43]. We interleaved fluorescence measurements with light reflectance measurements at 200 Hz, to obtain voltage and hemodynamic signals quasi-simultaneously. The hemodynamic measurements were subsequently used for offline correction for hemodynamic components of the fluorescence signals (see "Methods" section).

We observed spontaneously occurring bouts of 5-Hz oscillations in V1 (Fig. 1c, highlighted in yellow) during normal waking state while mice were exposed to an isoluminant blank screen. These bouts occurred at a median rate of 0.025 oscillatory episodes per second (mean = 0.032 Hz ± 0.046 standard deviation), consistent with the findings of Einstein et al., 2017[10]. In addition, we also binocularly presented a short-lasting (200 ms duration) moving grating visual stimulus, and detected 5-Hz oscillations that occurred time-locked to the visual stimulus presentation. Trial-by-trial time-resolved 5-Hz power (Fig. 1d) shows the occurrence of both spontaneously occurring and evoked 5-Hz bouts across imaging trials.

### 5-HT2AR agonist treatment increases the occurrence of spontaneous 5-Hz oscillations in V1, but not their power or duration

Next, to modulate cognitive function, we systemically applied a 5-HT2AR agonist (DOI or TCB2) half-way through an experimental session, generating a set of trials before and after injection (Fig. 2a).

We first examined the effect of 5-HT2AR agonist treatment on the spontaneously occurring 5-Hz oscillations. Time-resolved spectral analysis of V1 population voltage activity indicated the presence of both spontaneous and evoked 5-Hz oscillations in imaging sessions after agonist treatment (Fig. 2b; in this example using DOI). Compared to trials before injection, 5-HT2AR activation caused a significant increase in the rate of spontaneous 5-Hz oscillations (p < 0.001, Welch's t test, N = 78 (before), 87 (after) trials from 3 mice, Fig. 2c), with a median rate of 0.05 episodes/s. We observed no statistically significant increase in the power and duration (p-value = 0.885 and 0.654, respectively, Welch's t test, N = 55 (before) and 152 (after) oscillations) of these spontaneously occurring episodes of 5-Hz oscillations.

### 5-HT2AR agonist treatment increases the power and duration of visually evoked 5-Hz oscillations in V1

Next, we characterised the dynamics of the visually evoked oscillations. We therefore first separated all trials into two categories - those with evoked oscillations vs those without (Fig. 3ai, ii show mean traces of trials for each category, respectively; Fig. 3b depicts the corresponding wavelet analysis for trials before injection and with oscillations). Compared to the spontaneously occurring oscillations, visually evoked oscillations appeared similar in shape and power (p-value = 0.885, Welch's t test), and showed a statistically non-significant tendency of longer duration (median duration for spontaneous bouts: 540 ms and evoked bouts: 700 ms; p-value = 0.0579, Welch's t test). The across-trial average of evoked oscillations (Fig. 3ai) shows that each cycle of oscillations involved a phase of strong depolarization followed by hyperpolarization of the entire V1 pyramidal cell population. Figure 3aii depicts the mean of trials that showed no evoked bouts and only a single depolarisation-inhibition primary sensory responses were detected.

After injection of the 5-HT2AR agonist, we observed an increase in the duration and power of the evoked 5-Hz oscillations in V1 (Fig. 3aiii and Fig. 3c). These effects were consistent across all 7 experiments (Fig. 3d and Fig. 3e) and effective for either agonist used. TCB2 showed a significant increase in the power of the evoked oscillations (Supplementary Fig. 2a, **p-value = 1.09e-03, Welch's t-test) and a non-significant tendency to increase in duration (Supplementary Fig. 2b, p-value = 0.0796, Welch's t-test). The changes were not explained by other factors such as trial length or the number of imaging sessions carried out in the same mouse (Supplementary Fig. 3). A comparison between spontaneous and evoked oscillations after

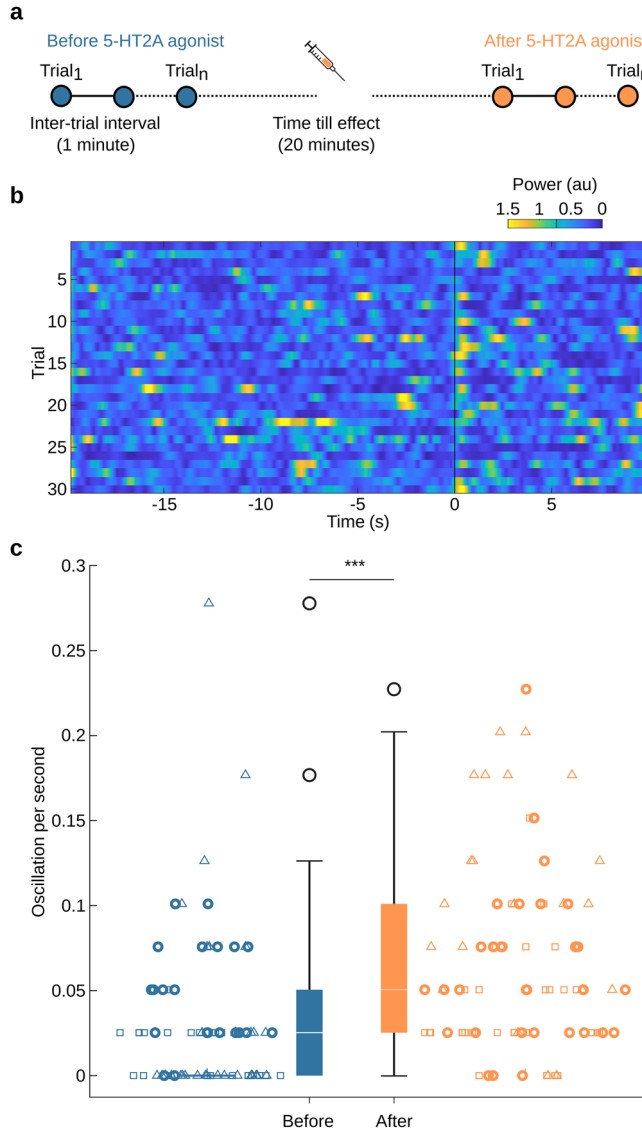

**Fig. 2 | The rate of spontaneous 5-Hz oscillations increases after injection of a 5-HT2A receptor agonist. a** Experimental timeline for trials before and after 5-HT2A receptor agonist injection. **b** 5-Hz time resolved spectral analysis of V1 membrane voltage for a single session after injection of DOI. The visual stimulus is presented at time zero (vertical black line). Vertical black line on the colour bar indicates the threshold for 5-Hz power counted as 5-Hz bout (see "Methods" section). **c** Quantification of spontaneous 5-Hz oscillations occurring in trials before (blue, $n = 80$) and after (orange, $n = 87$) injection, based on a subset of experiments with longer recordings prior to visual stimulation (20-s instead of 2 or 3-s in all other experiments) to capture extended periods of spontaneous activity (***$p$-value = 2.93e-05, Welch's $t$-test). $N = 3$ mice, data points representing each trial while symbol shape indicates individual mouse experiments; Boxplot follows Tukey's method: median (middle line), 25th, 75th percentile (box), ± 2.7 sigma (whiskers), outliers are plotted as 'o').

injection revealed a significant increase in the duration of the visually evoked 5-Hz oscillations (median duration for spontaneous bouts: 480 ms and evoked bouts: 1070 ms; $p$-value = 1.26e-9, Welch's t test).

### Evoked 5-Hz oscillations co-occur in V1 and the retrosplenial cortex (RSC)

Our widefield imaging approach covers the entire dorsal brain surface (Fig. 1b), so we next investigated whether visually evoked oscillations were present in other cortical regions. We only detected episodes of 5-Hz oscillations in the retrosplenial cortex (Fig. 4a, outlined in pink; RSC), which co-

occurred with oscillations in V1 (Fig. 4b and Supplementary video 1; Supplementary Fig. 4). We additionally examined several other cortical areas including another primary sensory modality (barrel field) and motor areas (primary and secondary motor areas), and we did not detect any notable occurrence of 5-Hz oscillations (Supplementary Fig. 4c, d). The correlation of V1 and RSC 5-Hz oscillations suggests a strong interactive function between these cortical areas[13]. In support of such specific relationship, we observed correlation in the duration of the oscillations between these two regions, which persisted after 5-HT2AR agonist application (Fig. 4c–e).

Notably, the oscillatory power between the two regions correlated only after injection of a 5-HT2AR agonist (Fig. 4e). This indicates that the 5-HT2AR agonist strengthened the interactions between V1 and RSC. To explore whether this interaction was direct or whether it resulted from a common source, we quantified the lag between the two signals. The cross-correlation of co-occurring V1 and RSC oscillations revealed a consistent mean lag for both before and after injection of the agonist, of -18.6 ms and -17.7 ms respectively (Fig. 4c; a similar result was found using FFT analysis to calculate the phase difference, Supplementary Fig. 5).

### 5-HT2AR agonist treatment increased the occurrence probability of 5-Hz oscillations

We found that 5-HT2AR activation enhanced the occurrence probability of both spontaneous and stimulus-evoked 5-Hz oscillations in V1 and RSC in a similar manner (Fig. 5a, b). As in V1, we also see a significant increase in the rate of spontaneous oscillations occurring in the RSC after agonist treatment (Fig. 5b; before: mean = 0.085 Hz +/- 0.093, after: mean = 0.132 Hz +/- 0.086). TCB2 produced a more pronounced increase in probability for evoked oscillations compared to DOI (Supplementary Fig. 2c).

Further, we found that evoked 5-Hz oscillations predominantly either co-occurred in both regions or in V1 alone, but rarely in RSC alone (Fig. 5c). This observation is maintained after agonist treatment. The rare occasion of an evoked oscillation only in RSC was also combined with power and duration on the border of detection (Supplementary Fig. 6a, b), suggesting spurious RSC-only positives and further lending weight to the interpretation of a V1-driven effect upon visual stimulus.

Overall, the increase of visually evoked oscillations in V1 and RSC suggests an enhanced recruitment of underlying processes such as memory retrieval[44], stimulus interpretation, and prediction.

### Discussion

Cortical oscillations can modulate perception and have been linked to various cognitive states and functions[1–4]. While psychedelics are also generally known to alter perception and states of consciousness, whether these serotonergic hallucinogens modulate oscillatory activities in the cortex remains largely unknown. Here, we measured neuronal population voltage signals of pyramidal neurons across the entire mouse dorsal cortex at high spatiotemporal resolutions and coverage using epifluorescence widefield imaging in awake head-fixed mice. We detected episodes of 5-Hz oscillations in the primary visual cortex (V1), occurring both spontaneously and evoked by presenting a visual stimulus. We observed experimental evidence for how serotonergic psychedelics modulate oscillatory events, revealing a possible correlate of the altered brain-state. Expanding beyond V1, we also observed 5-Hz oscillatory activities in the retrosplenial cortex (RSC). We further obtained experimental measures of how inter-areal cortical activities between V1 and RSC are altered by psychedelics, possibly through increasing interregional connectivity strengths.

Our focus on oscillations in the theta range was driven by its implicated role in many cognitive functions, including cortical information processing[45], working memory[8], memory encoding and retrieval[10–13]. The existence of 5-Hz oscillations in V1 has previously been reported and their occurrence has been strongly correlated to the degree of familiarity to a given visual stimulus[10–12]. Specifically, others have reported that V1 5-Hz oscillations displayed an increase in duration, power, and probability of occurrence when mice perceived a visual stimulus repeatedly presented over consecutive days[11,12]. Our study does not provide additional evidence for

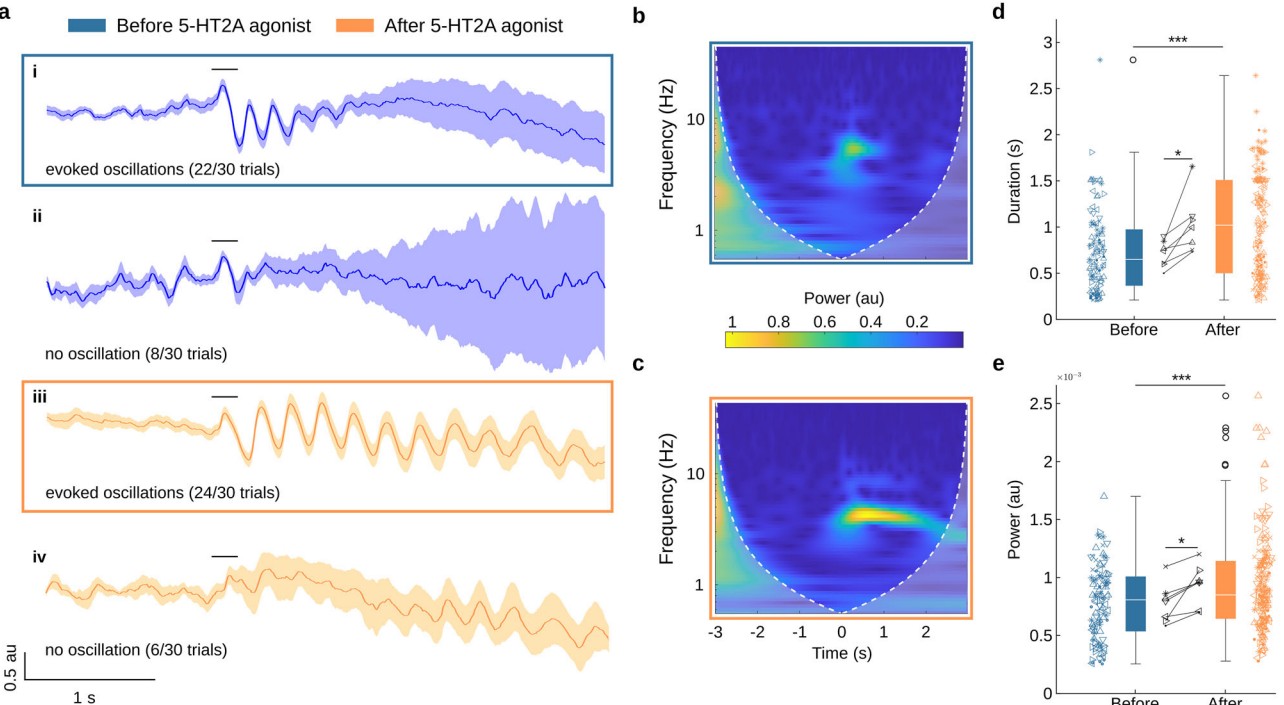

**Fig. 3 | Visually evoked 5-Hz oscillations in V1 and their dynamic changes following treatment with 5-HT2AR agonist. a** Trial averaged V1 population membrane voltage activity (shading represents SEM) before (i-ii) and after (iii-iv) treatment of the 5-HT2AR agonist (TCB2 in this example). **b** Time-resolved spectral analysis of the evoked oscillations shown in **a**-i. Time zero indicates onset of visual stimulus. **c** Same as **b** after drug treatment (**a**-iii), showing an increase in both duration and amplitude of 5-Hz oscillations. **d** Duration of oscillations before and

after treatment comparing the mean across all experiments (indicated inside, *$p$-value = 0.0144, paired $t$ test) and individual trials (indicated outside, ***$p$-value = 1.35e-06, Welch's t-test, $n_{before}$ = 109 and $n_{after}$ = 189). Data based upon $N = 7$ experiments in 5 mice, symbol shape indicates individual mouse experiments. **e** Same as **d**, for mean oscillatory power across all experiments (indicated inside, *$p$-value = 0.016, paired $t$ test) and individual trials (indicated outside, ***$p$-value = 5.91e-04, Welch's t-test).

5-Hz oscillations in V1 relating to perceptual familiarity, as we do not have a measure of familiarity beyond the trial number. Also, our data did not show an increased occurrence of 5-Hz oscillations with increasing trial number.

Our experimental paradigm involved the passive viewing of a visual stimulus rather than behavioral task conditions where a visual stimulus may have an attached valence that could influence its processing. Hence, the observed oscillations could likewise reflect attentional disengagement or distraction from repetitive stimuli[46,47]. While we cannot rule out this possibility, under the assumption that an animal is likely to be more attentive and engaging at the beginning of an imaging session than at the end, we did not observe any changes of 5-Hz oscillations across trials of a given imaging session. Therefore, making this possibility unlikely, nevertheless, future studies may empirically test this possibility.

Psychedelics are primarily known to influence visual processing through activation of 5-HT2A receptors[29,48]. We observed 5-HT2A receptor agonists (DOI and TCB-2) facilitate significant modulation on bouts of 5-Hz oscillations in V1 and the RSC. First, we considered these observations in the context of psychedelics' capacity to alter the brain state and the reported possibility of pupil dilation induced by the acute effect of psychedelic administration[49]. There is accumulating evidence which supports that cortical state changes are intricately interwoven with interoceptive signals[50,51] and visual responses[52]. Moreover, recent studies suggest theta-range rhythms may emerge from sensorimotor sampling frequency[53,54]. However, though psychedelics may induce pupil dilation[49], recent studies in humans suggest that N,N-dimethyltryptamine (DMT, a 5-HT2AR serotonergic psychedelic[55] similar to those used in this study) can modulate V1 population receptive field sizes devoid of motor-related signals from eye or head movements[56]. In fact, in mice, previous observations by others have shown a lack of changes in both pupil diameter and in general behavioural state following administration of DOI[24] (as used in the current study). In line with this, we did not observe any correlation in our data between properties

of the 5-Hz oscillation and trial number after drug injection. This led us to conclude that the observed psychedelic-induced changes on 5-Hz oscillations were not due to changes in pupillometry or in the animals' behavioural state. Subsequently, considering the effects induced by the systemic injection itself, we found that properties of V1 5-Hz oscillations were altered by both types of 5-HT2A agonists used, although significantly modulated more by TCB2 than by DOI (Supplementary Fig. 2). This drug dependency makes a pure stress-induced artifact originating from the injection procedure unlikely. Taken together, this led us to consider the alterations in cortical 5-Hz oscillations as a physiological effect of the psychedelic agents.

Capitalizing on the spatial coverage of our approach, we expanded our investigation to beyond the visual cortex and across both cortical hemispheres. We found that visually-evoked oscillations in V1 frequently co-occurred with similar bouts in the RSC, a major hub connecting hippocampus with the cerebral cortex. Bouts of 5-Hz oscillations occur in both V1 and in the RSC with a delay of ~18 ms. Given a distance of 1.5–3.2 mm (Supplementary Fig. 1e) between V1 and RSC, this translates to a propagation speed of 0.083–0. 12 m/s, a value in the range of traveling waves in the cortices of the mouse (0.08–0.15 ms⁻¹[57,58]), cat (0.1–0.3 ms⁻¹[59–61]), monkey (0.25–1.35 ms⁻¹[62,63]) and human (0.3–0.5 ms⁻¹[64]). Also, the range of calculated propagation speed aligns with those found for unmyelinated (long-range) axonal conduction speeds within superficial cortical layers[65], which suggests that V1 occasionally triggers oscillations in RSC upon visual input. This is further supported by the fact that visually-evoked oscillations in the RSC were rarely found alone.

We observed no notable co-occurrence of 5-Hz oscillations between V1 and other cortical regions, other than the RSC, including cortical areas for other sensory modalities (e.g., the barrel field cortex) or motor areas (primary and secondary motor cortices). The spatially restricted existence of these 5-Hz oscillations strongly suggests their specific involvement in the processing of visual information, extending importance to the RSC which is

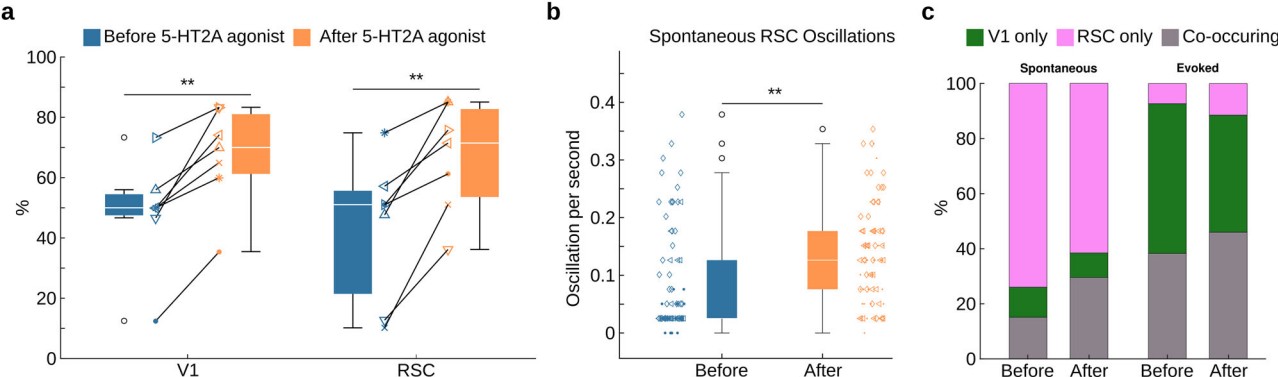

**Fig. 4 | Visually-evoked V1 5-Hz oscillations are followed by similar oscillations in the retrosplenial cortex. a** A series of single trial cortex-wide voltage maps of visual stimulus-evoked oscillation of population membrane voltage in both V1 and RSC after TCB2 injection (Supplementary video 1 as an additional example for data shown in Fig. 3a, b). **b** Changes in membrane voltage averaged across V1 and RSC (regions marked in **a**) following visual stimulus presentation (black bar). **c** Quantification of the lag time (calculated by cross correlation) between signals (based on a subset with high power) in V1 and RSC across single trials before (blue,

$n$ = 28) and after injection (orange, $n$ = 59). Cross-correlation plots are shown in Supplementary Fig. 5b–d. **d** Linear regression and correlation between the duration of evoked 5-Hz oscillations in V1 and RSC, both before (left; $R^2$ = 0.95 and \*\**p*-value = 0.00189, $n$ = 64) and after injection of 5-HT2A agonist (right; $R^2$ = 0.695 and \*\**p*-value = 0.00217, $n$ = 158). **e** Same as **d** for oscillation power before (left; $R^2$ = 0.71 and n.s, *p*-value = 0.718) and after injection of 5-HT2A agonist (right, $R^2$ = 0.692 and \*\*\**p*-value = 6.28e-04). (All R-values were calculated using bisquare weights and *p*-values depict the Pearson's correlation.).

**Fig. 5 | 5-HT2AR agonist increased the occurrence probability for 5-Hz oscillations in both V1 and RSC. a** Wavelet analysis detected evoked 5-Hz oscillations in V1 (\*\**p*-value = 0.002, paired T test) and RSC (\*\**p*-value = 0.00275, paired T test) in an increased percentage of trials after 5-HT2AR agonist treatment (*N* = 7 experiments in 5 mice). **b** Increase in frequency of spontaneous 5-Hz oscillations in RSC by

5-HT2A agonist. (\*\*\**p*-value = 1.26e-04, Welch's t-test. *N* = 3 mice, individual data points represent each trial, data point shape represents the individual mouse. $n_{before}$ = 75 and $n_{after}$ = 84). **c** Distribution of spontaneous (left) and evoked (right) oscillations occurring only in V1, RSC or co-occurring in both regions (see Methods).

increasingly implicated in integrating internal representations with visual input[66].

Oscillatory activity around 5 Hz has been observed in subcortical structures, including the thalamus and hippocampus, which in some instances appear to co-occur with oscillations in V1. Seminal work of McCormick and colleagues have tied V1 oscillatory activity in the 3–5 Hz range to the thalamocortical loop, specifically, when animals were engaged in active sensing associated with arousal and movements[9]. More recently, however, under passive viewing conditions paired with extracellular recordings have shown that V1 oscillations do not originate in the thalamus[13]. That is, inactivation of the hippocampus did not affect V1 oscillations, indicating that these oscillations emerge independent of each other[13] and contingent on behavioural context. The independent emergence of these oscillations may contribute to the seeming discrepancy between the facilitatory effects on cortical 5-Hz oscillations by serotonergic psychedelics (i.e., 5-HT2A agonists) seen here vs the depression in the power of similar oscillations in the hippocampus induced by serotonin[21,22] acting on different 5-HTR simultaneously[13].

While the cellular and circuit mechanisms of cortical 5-Hz oscillations remain to be elucidated, 5-Hz oscillations could in principle emerge from a strengthening of local circuitries within V1 (see ref. 5 for a recent review), in particular, when exposed to a specific familiar stimulus[13]. Interestingly, familiarity-evoked theta oscillations were longer and less stimulus specific in SERT knockout (KO) mice[14]. On a mechanistic level, activation of the 5-HT2A receptor by specific agonists affects pyramidal cell dendritic excitability and interneuron interactions and could for example modulate slow oscillations via somatostatin-expressing neurons[36].

We found a strong facilitatory effect of 5-HT2A agonists on spontaneous and visually-evoked 5-Hz oscillations within both V1 and RSC. Following changes to the cortical state through systemic activation of such a single receptor type, the detected enhancement in V1–RSC interaction may be interpreted as a "cascade-effect" of bouts in V1 on bouts in the RSC[67,68]. This supports the concept that brain states under the influence of hallucinogenic agents increase internally driven and/or decrease externally driven activity, resulting in visual perceptions being under increased top-down control[69–71] where they are overly guided by internally stored or acutely generated associations.

## Materials and methods
*Key Resources*

| REAGENT or RESOURCE Experimental Models: Organisms/Strains | Source |
|---|---|
| Mouse: Ai87XAi93 | 42,43 |
| Mouse: CamK-tTA;chiVSFP | 72,73 |
| **Chemicals, Peptides, and Recombinant Proteins** | |
| ( ± )-DOI hydrochloride | Sigma-Aldrich, Cat#: D101-10MG |
| TCB-2 | Tocris Bioscience, Cat#: 2592/10 |

## Animals
We have complied with all relevant ethical regulations for animal use. All procedures were conducted in accordance with the guidelines of with the European Union Community Council guidelines and approved by the German Animal Care and Use Committee under the Deutsches Tierschutzgesetz (Az.: 84-02.04.2019.A477, 84-02.04.2019.A483) and the NIH guidelines. The experiments were carried out on 5 (4 male, 1 female) mice, aged 4-5 months, using strains Ai87XAi93 ($N = 2$) and CamK-tTA;chiVSFP

($N = 3$), expressing voltage-indicators in layer II/III or in layer II/III plus layer V pyramidal neurons, respectively. After preparatory surgery of the cortical window, mice were housed in standard vivarium conditions (temperature maintained at 20–22 °C, 30–70% humidity) and kept in 12 h light/dark cycle with food and water ad libitum.

### Surgical procedure
Surgery was carried out as previously described[23,74]. In brief, animals were anesthetized with isoflurane (3% induction and 1.5–2% for maintenance via a nose mask) and received a subcutaneous bolus of isotonic 0.9% NaCl solution mixed with Buprenorphine (0.1 µg/g bodyweight) and Atropine (0.05 µg/g bodyweight). A heating pad (set to 37 °C) was placed below the animal during surgery to maintain body temperature. Before sagittal incision along the midline, 2% Lidocaine was applied to provide additional local anaesthesia. The skull was slowly thinned using a microdrill until the surface blood vessels became clearly visible.

The thinned skull was then covered with transparent dental cement and a glass cover for protection and to reduce light scattering. Finally, a custom head holder was attached to the skull in order to provide a clear and easily accessible imaging window for chronic experiments. Mice were allowed to recover for at least one week before beginning habituation to the imaging setup.

### Voltage imaging
Simultaneous recording of Fluorescence at 515–569 nm and >594 nm as well as reflectance recording at 515–569 nm and 620–660 nm was performed using interlaced imaging with different light sources switching every 5 ms. The fluorescence signals (FRET donor and acceptor) carry a combination of the voltage and hemodynamic signals while the reflectance signals carry only the hemodynamic signals (blood volume and blood oxygenation). The fluorescence signals were then modelled as a linear combination of the reflectance (hemodynamic) signals and simple regressors. Using this model, it was possible to remove the hemodynamic component from the fluorescence signal, extracting the voltage signal.

### Visual stimulus
A monitor (60 Hz, mean luminance 55 cd/m2, LG 24BK55WV-B) was placed 30 cm away from the stimulated eye, covering ~56 × 70 degrees of the visual field. A semipermeable zero power contact lens was used to prevent the eyes from drying out. The visual stimuli consisted of a moving grating with a 90 degree orientation, 100% contrast and a duration of either 0.2 or 0.5 s. This was followed by a uniform grey screen (blank condition) presented for 15 s during the interstimulus intervals. Each experiment consisted of 30–110 trials without and then with treatment of the 5-HT agonist.

### 5-HT2AR agonist treatment
TCB-2 (Tocris Bioscience, 10 mg/kg bodyweight in saline) or DOI (Sigma, 10 mg/kg bodyweight in saline) was administered subcutaneously[24] and after waiting ~20 minutes (time till effect) recordings continued with the same experimental protocol as before. The dosage was chosen to meet literature standards of intraperitoneal injections[24,31,32,68,75,76]. As mice were head fixed, injections were performed without removing the animal from the setting to stabilise the transition between conditions.

### Data analysis
Voltage imaging data was preprocessed and functionally aligned to the Allen Mouse Brain Atlas projected upon the imaging plane (Fig. 1b, Supplementary Fig. 1), as previously described[73].

All data analysis was performed in MATLAB (R2024b Update 2), unless stated otherwise all functions refer to MATLAB built-in functions. For the analysis of the membrane potential ($V_m$), 5-Hz oscillations were detected by wavelet analysis (using the *cwt* function). For each mouse the right hemisphere was analyzed. The 5-Hz oscillations were identified by those traces which had a 5-Hz magnitude (calculated using the mean of the 4.8–5.2 Hz band) 4 standard deviations above the baseline (pre-

stimulus) for a duration of at least 0.2 s (one 5-Hz oscillation). Detection was also verified by manual detection (using a custom Python 3.12.7 script GUI) and Fast Fourier transform (using the *fft* function. Evoked oscillations were classified as co-occurring if oscillations time-locked to the visual stimulus were identified in both V1 and RSC, further verified by calculating the time lag between the signals. The time lag was calculated by cross correlation (using the *xcorr* function) and then compared to FFT lag time. For FFT, the signals were transformed into the frequency domain (using the *fft* function) and then calculated the phase (using the *angle* function) of the closest bin to 5 Hz. The subsequent phase difference between the signals was then transformed into time lag by dividing the angular frequency ($\omega = 2\pi f = 2\pi 5$).

Spontaneous oscillations were identified using the same thresholding method as evoked (without the time-locked restriction), creating a binarized vector of on-going activity and subsequently testing co-occurrence if V1 and RSC oscillations overlapped by at least one image frame.

## Statistics and reproducibility

Statistical analysis was performed using MATLAB (R2024b Update 2), unless stated otherwise all functions refer to MATLAB built-in functions. All statistical analysis was performed using a two-sided independent T-test (using the ttest2 function). Boxplots used follow Tukey's method: median (middle line), 25th, 75th percentile (box), ± 2.7 sigma (whiskers), outliers are plotted as an outlined 'o'. For comparison of uneven sample sizes, we used Welch's T-test (using the ttest2 function with "Vartype" settings to "unequal"). While experiment-wise means before and after injection of the agonist were compared using a paired T-test (using the ttest function). We calculated the linear regression (using the fit function with settings "poly1" and "bisquare" robustness) between V1 and RSC, while *p*-values represented the Pearson correlation (using the corr function).

The standard identification of statistical significance was used: \*$p < 0.05$, \*\*$p < 0.01$, \*\*\*$p < 0.001$.

## Data availability

The data supporting the findings of this study and custom MATLAB code used for analysis are available within the main text, the Supplementary Information file, the Supplementary Data, or from the corresponding authors upon request.

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

## Acknowledgements

We thank Stefan Dobers and Henning Knoop and the RUB mechanical shop for technical support. This work was supported by Deutsche Forschungsgemeinschaft (DFG) grants: Project ID 122679504 - SFB 874, D.J.; JA 945/5-1, D.J.; "MoNN&Di", Project number 492434978 - GRK 2862/1, Subproject 10, D.J.; BMBF, ERA-Net Neuron "Horizon 2020", 01EW2104B, D.J.; US National Institutes of Health BRAIN Initiative Grant (5U01NS099573), T.K.; Lee Kuan Yew Postdoctoral Fellowship administered by Nanyang Technological University Singapore (022506-00001), C.S.; Open Fund Young Individual Research Grant (MOH-001720) administered by the Singapore Ministry of Health's National Medical Research Council, C.S.

## Author contributions

These authors contributed equally: Zohre Azimi, Robert Staadt and Callum M. White. These authors jointly supervised this work: Dirk Jancke and Thomas Knöpfel. Conceptualization: T.K. and D.J. Data curation: C.M.W. and D.J. Formal analysis: C.M.W. and D.J. Data acquisition: Z.A. and R.S. Data visualization: C.M.W. and D.J. Software: C.M.W., T.K. and R.S. Supervision: T.K. and D.J. Funding acquisition and resources: C.S. T.K. and D.J. Writing—original draft: C.M.W., C.S., T.K. and D.J.

## Funding

## Competing interests

The authors declare no competing interests.
