## [Transparent Peer Review file · Communications Biology]

Psychedelic 5-HT_{2A} agonist increases spontaneous and evoked 5-Hz oscillations in visual and retrosplenial cortex

Corresponding Author: Professor Dirk Jancke

Version 0:

Reviewer comments:

Reviewer #1

(Remarks to the Author)

The paper is focused on the mechanism of spontaneous 5-Hz oscillations in the visual and cortex of awake mice. Low frequency oscillations are known to modulate visual perception and are hypothesized to be involved in rhythmic sampling of information in sensory cortices, working memory, modulation of attention and memory consolidation. Serotonin (5-HT) is known to modulate the strength of evoked activity in the visual cortex, but how it effects low frequency oscillations is not well studied. The authors use genetically encoded voltage indicators (GEVIs) in combination with full brain mesoscale voltage imaging of V1 and RSC and discover both visually triggered and spontaneous 5 Hz oscillations of membrane potential with RSC following V1. Injection of 5-HT_{2A} agonist increases the rate, but not the power or duration of spontaneous oscillations. However, it does increase the rate, power and duration of visually evoked 5 Hz oscillations.

The paper is very strong and is very technically innovative. Mesoscale imaging of voltage signals at 200 Hz acquisition rate across the whole brain from multiple cortical areas is outstanding. The finding of spontaneous oscillations is also very interesting. The interaction between V1 and RSC is important. The findings of the paper, including the effect of 5-HT_{2A} agonist on the 5 Hz oscillations both in V1 and RSC will bring more insight into the neural circuit mechanism of sensory perception. The connection between 5 Hz oscillations, serotonergic agonists and hallucinations is intriguing.

Overall, the paper is strong and innovative, it is technically sound with different analyses of the oscillations in V1 and RSC and their interaction, including membrane voltage activity, spectral analysis.

There are some questions left:

- 1) How repeatable and consistent trial-to-trial are the oscillations in V1 and RSC?
- 2) It would be interesting to see examples of cross-correlations of the V1 and RSC oscillations in Figure 4, not just the lag time
- 3) The large differences between spontaneous oscillations in RSC vs V1 and evoked oscillations in V1 vs RSC are fascinating and should probably be discussed more.
- 4) It would also be useful to do functional connectivity analysis of V1 and RSC for both the spontaneous and evoked oscillations, for example a Granger causality analysis.
- 5) The connection of these 5 Hz oscillations to hallucinations may be fascinating, but this connection needs to be very carefully discussed. Perhaps it would be useful to expand the related literature relating 5 Hz oscillations, serotonin system and hallucinations.

Minor issues:

- 1) Line 21 "... a visual stimuli";
- 2) Line 325 "... (REF michiael)...";
- 3) The statistics description is too brief.

Reviewer #2

(Remarks to the Author)

In this manuscript, White and colleagues explore the initiation and propagation of ~5 Hz oscillations across the cortex in awake mice, using wide-field voltage imaging. Prior studies, particularly from the Golshani and McCormick laboratories, have characterized similar low-frequency, alpha-like rhythms (~3–5 Hz) in the visual cortex, demonstrating that they suppress sensory-evoked responses and involve thalamocortical interactions. This body of work supports the notion that

such oscillations may represent an analogue of the alpha rhythm in mice. However, the foundational publications from the McCormick lab are not cited here and should be acknowledged to properly situate the current study in the context of existing literature.

The authors leverage the advantages of voltage imaging to investigate these oscillations' origins and dynamics. Their results suggest that initiation occurs in the visual cortex, followed by propagation most prominently to the retrosplenial cortex. Additionally, they present compelling evidence that administration of 5-HT_{2A} receptor agonists enhances both the probability of initiating these oscillations and their duration, a finding that aligns with previous literature suggesting oscillation generation is state-dependent and most prevalent during quiet, awake resting state.

The experimental design is well conceived, and the results are convincing. However, I believe the analysis could be strengthened in several key areas:

1. Expanded Spatiotemporal Analysis

Currently, the analysis focuses mainly on the visual and retrosplenial cortices. A more comprehensive mapping of the initiation and propagation patterns, especially to other cortical areas such as frontal regions, would leverage the strength of functional voltage imaging and yield far more novel insights as we currently don't know where those oscillations originate and if the propagation pattern is stereotypical. The current emphasis on retrosplenial cortex appears somewhat arbitrary without broader context.

2. Control for Injection-Related Effects

Though stress from injections is unlikely to account for the observed changes in oscillations, a saline-injected control group would serve as a critical negative control to rule out non-specific effects of injection procedures.

3. Discussion of Thalamocortical Contributions

The discussion should engage more directly with the possibility that these 5 Hz oscillations involve thalamocortical loops. Prior studies have shown that alpha-like rhythms (~3–5 Hz) in V1 depend on thalamic activity, with silencing of the visual thalamus abolishing the rhythm. Delving deeper into this mechanism would strengthen mechanistic interpretation. Hence, this study represents a significant advance; incorporating broader spatiotemporal analyses, appropriate controls, and deeper mechanistic discussion will substantially improve its impact and rigor.

Reviewer #3

(Remarks to the Author)

This study presents compelling evidence that systemic administration of serotonin 5-HT_{2A} receptor agonists enhances both spontaneous and visually evoked 5-Hz oscillations in V1 and RSC of awake mice. By leveraging cortex-wide voltage imaging, the authors provide a rare mesoscale perspective on how hallucinogens modulate inter-areal cortical dynamics. The observation of strengthened V1–RSC coupling under psychedelic treatment is particularly novel and potentially impactful for our understanding of the neural basis of perception and posterior cortical oscillatory states. Their technique is really impressive. This represents a potentially important contribution to systems neuroscience. However, several points merit further clarification and discussion.

1. Interpretation of 5-Hz rhythms in the absence of task engagement

The current paradigm relies on passive viewing, with no behavioral task or valence associated with the visual stimuli. Previous studies have reported that ~5 Hz activity in visual cortex is particularly prominent when animals disengage from visual input (Spyropoulos et al., PNAS, 2018; Han et al., eNeuro, 2019). Thus, the observed oscillations could reflect not familiarity per se, but rather an attentional disengagement or "ignoring" of repetitive stimuli. It remains unclear whether the mice were actively attending to the monitor or whether the stimuli were relegated to peripheral vision. The authors should cite these references and explicitly discuss whether passive viewing may have facilitated a state in which oscillations were enhanced by inattention rather than stimulus familiarity.

2. Behavioral phenotypes of systemic serotonin agonists

SC administration of the drug is well documented to induce hyperactivity, stereotypy, and head-twitch responses in rodents. These behavioral phenotypes suggest a broad alteration of brain and bodily states, beyond local cortical modulation. The authors should acknowledge this literature and consider how systemic behavioral effects could interact with, or confound, the observed cortical oscillations. For example, the drugs are assumed to cross the BBB, could the observed effects alternatively reflect peripheral (e.g., muscular or ocular) actions that secondarily altered behavioral or brain state?

3. Retinotopic coverage of the stimulus

The visual stimuli used covered ~50-70° of visual angle, while the mouse visual field extends beyond 260°. Thus, a substantial portion of the mouse's visual field remained unstimulated. If animals were predominantly processing the stimuli in their peripheral vision, the functional interpretation of the evoked oscillations might differ from what would be observed under full-field or behaviorally salient conditions. Discussion of this limitation, and how it constrains interpretation of "whole V1" dynamics, would strengthen the manuscript.

4. Spatiotemporal patterns of oscillatory propagation

The supplemental videos reveal impressive traveling wave-like propagation of oscillations. A more quantitative analysis of spatiotemporal phase relationships, perhaps using statistical parametric mapping (SPM) across the cortex, would greatly enrich the characterization. This would also clarify whether frontal/motor regions show consistent ~5 Hz phase dynamics relative to V1 and RSC. In short, what is the spatial/spatio-temporal span of 5 Hz rhythm?

5. Functional origin of the oscillations

The authors report facilitation of 5-Hz oscillations, in contrast to hippocampal theta where serotonin typically suppresses its

power. This discrepancy deserves further discussion. Why should visual oscillations be amplified while hippocampal theta is dampened? If the oscillations are not stimulus-driven, what is their functional role? Recent studies suggest theta-range rhythms may emerge from sensorimotor sampling frequency (Joshi et al., *Nature*, 2023; Forli et al., *Nature*, 2025; Bagur et al., *Nat Comm*, 2021; Karalis & Sirota, *Nat Comm*, 2022). In visual domain, in humans, inter-saccadic intervals cluster around ~200 ms (~5 Hz; Otero-Millan et al., *J Vision*, 2008). Could ocular drift or microsaccades induced by the stimulus be contributing to the observed rhythmicity? Clarifying the functional and physiological origin of these oscillations would strengthen the interpretation. I believe the origin of the rhythm is endogenous, but I believe future readers would wonder what is the role and origin of the rhythm.

6. Network mechanism of oscillation generation

From a network-model perspective, it would be valuable to elaborate on the putative circuit mechanisms. 5-HT_{2A} activation is known to modulate pyramidal cell dendritic excitability and interneuron interactions, which modulates slow oscillations (de Filippo et al., *eLife*, 2021). At the same time, recent reports indicate that ~5 Hz rhythms in visual cortex can originate locally within V1/V4 (Zimmerman et al., *Curr Biol*, 2025; Kienitz et al., *Curr Biol*, 2021). A discussion integrating these findings—whether the rhythm reflects a pyramidal–interneuron loop, a dendritic resonance effect, PING- or ING-based oscillator circuits, a cascade into RSC, or a rhythmic outside source—would provide important mechanistic context.

Version 1:

Reviewer comments:

Reviewer #3

(Remarks to the Author)

Authors have addressed all of my concerns. The revised manuscript looks ready for publication.

Point-by-point response to reviewer comments

We thank all three reviewers for their helpful comments on our manuscript and we are grateful for their suggestions for improvement. Please see below our point-by-point reply to the raised comments and the revised manuscript with all changes marked in red.

REVIEWER COMMENTS

Reviewer #1 (Remarks to the Author):

The paper is focused on the mechanism of spontaneous 5-Hz oscillations in the visual and cortex of awake mice. Low frequency oscillations are known to modulate visual perception and are hypothesized to be involved in rhythmic sampling of information in sensory cortices, working memory, modulation of attention and memory consolidation. Serotonin (5-HT) is known to modulate the strength of evoked activity in the visual cortex, but how it effects low frequency oscillations is not well studied. The authors use genetically encoded voltage indicators (GEVIs) in combination with full brain mesoscale voltage imaging of V1 and RSC and discover both visually triggered and spontaneous 5 Hz oscillations of membrane potential with RSC following V1. Injection of 5-HT_{2A}R agonist increases the rate, but not the power or duration of spontaneous oscillations. However, it does increase the rate, power and duration of visually evoked 5 Hz oscillations.

The paper is very strong and is very technically innovative. Mesoscale imaging of voltage signals at 200 Hz acquisition rate across the whole brain from multiple cortical areas is outstanding. The finding of spontaneous oscillations is also very interesting. The interaction between V1 and RSC is important. The findings of the paper, including the effect of 5-HT_{2A}R agonist on the 5 Hz oscillations both in V1 and RSC will bring more insight into the neural circuit mechanism of sensory perception. The connection between 5 Hz oscillations, serotonergic agonists and hallucinations is intriguing.

Overall, the paper is strong and innovative, it is technically sound with different analyses of the oscillations in V1 and RSC and their interaction, including membrane voltage activity, spectral analysis.

We thank the reviewer for the kind support of our manuscript.

There are some questions left:

1) How repeatable and consistent trial-to-trial are the oscillations in V1 and RSC?

We agree this is an interesting point. We have now added mean values \pm std to give an indication of trial-to-trial variability.

2) It would be interesting to see examples of cross-correlations of the V1 and RSC oscillations in Figure 4, not just the lag time

We agree. We have now calculated the cross-correlation as the reviewer suggests:

Figure R1. Cross-correlations of the V1 and RSC oscillations.

We have added this as Supplementary Figure 5b-d in the revised manuscript.

3) The large differences between spontaneous oscillations in RSC vs V1 and evoked oscillations in V1 vs RSC are fascinating and should probably be discussed more.

We agree. We have expanded the discussion in the revised manuscript to discuss this observation.

4) It would also be useful to do functional connectivity analysis of V1 and RSC for both the spontaneous and evoked oscillations, for example a Granger causality analysis.

We appreciate where the reviewer is coming from and we agree that functional connectivity between V1 and RSC in the context of these 5-Hz oscillations may be interesting to explore. However, this current manuscript does not make claims regarding causality. We duly considered the Granger causality analysis that the reviewer suggested but we believe this analysis would not be suitable to the experimental approach at hand (e.g. we cannot exclude contribution of causal inputs from areas outside our imaging field of view of the dorsal cortex), and such causal functional analysis would better constitute a future study that is specifically designed to interrogate this.

Nevertheless, we have revised our discussion section to reflect this in the revised manuscript.

5) The connection of these 5 Hz oscillations to hallucinations may be fascinating, but this connection needs to be very carefully discussed. Perhaps it would be useful to expand the related literature relating 5-Hz oscillations, serotonin system and hallucinations.

We thank the reviewer for this suggestion and revised our discussion.

Minor issues:

1) Line 21 "... a visual stimuli";

Corrected, thank you.

2) Line 325 "... (REF michiael)...";

Corrected, thank you.

3) The statistics description is too brief.

We have now expanded this section, thank you.

Reviewer #2 (Remarks to the Author):

In this manuscript, White and colleagues explore the initiation and propagation of ~5 Hz oscillations across the cortex in awake mice, using wide-field voltage imaging. Prior studies, particularly from the Golshani and McCormick laboratories, have characterized similar low-frequency, alpha-like rhythms (~3–5 Hz) in the visual cortex, demonstrating that they suppress sensory-evoked responses and involve thalamocortical interactions. This body of work supports the notion that such oscillations may represent an analogue of the alpha rhythm in mice. However, the foundational publications from the McCormick lab are not cited here and should be acknowledged to properly situate the current study in the context of existing literature.

We thank the reviewer for this point and apologize for having overlooked these citations which are indeed relevant to our topic. We have added them into the introduction and discussion sections in the revised manuscript. In line with Golshani and McCormick we are now more specifically referring to 5Hz oscillations, as delta and alpha bands have been derived from human EEG and this division of the frequency spectrum does not directly apply to mice. We use the terms delta and alpha-like when referring to literature using these terms.

The authors leverage the advantages of voltage imaging to investigate these oscillations' origins and dynamics. Their results suggest that initiation occurs in the visual cortex, followed by propagation most prominently to the retrosplenial cortex. Additionally, they present compelling evidence that administration of 5-HT_{2A} receptor agonists enhances both the probability of initiating these oscillations and their duration, a finding that aligns with previous literature suggesting oscillation generation is state-dependent and most prevalent during quiet, awake resting state. The experimental design is well conceived, and the results are convincing. However, I believe the analysis could be strengthened in several key areas:

1. Expanded Spatiotemporal Analysis

Currently, the analysis focuses mainly on the visual and retrosplenial cortices. A more comprehensive mapping of the initiation and propagation patterns, especially to other cortical areas such as frontal regions, would leverage the strength of functional voltage imaging and yield far more novel insights as we currently don't know where those oscillations originate and if the propagation pattern is stereotypical. The current emphasis on retrosplenial cortex appears somewhat arbitrary without broader context.

The reviewer makes a very good point and indeed we were somewhat surprised to have not found other cortical areas with strong 5-Hz oscillatory activity. But we agree with the reviewer that our report of this observation in our initial submission could be further documented. Therefore, we now analyzed the 5-Hz power pixel-wise across several additional defined areas across the cortex, including additional sensory region (barrel cortex) and motor regions (primary and secondary motor cortical regions).

Our previous reported observations remain the same: we did not find significant 5-Hz oscillations as strong as that found in V1 and RSC.

Figure R2. Analysis of 5-Hz oscillations in other cortical areas.

This new analysis is provided in the manuscript as new Supplementary Figure 4 c-d.

2. Control for Injection-Related Effects

Though stress from injections is unlikely to account for the observed changes in oscillations, a saline-injected control group would serve as a critical negative control to rule out non-specific effects of injection procedures.

As the reviewer pointed out, stress from a subcutaneous injection is unlikely to account for the observed effects. Importantly, our reported effects of cortical 5-Hz oscillations were dependent on the type of drug used - those induced by DOI was less than that of TCB2, shown in Supplementary Figure 2c. This drug dependency indicates that that it is unlikely that our observations were stress-induced artifacts from systemic injections.

3. Discussion of Thalamocortical Contributions

The discussion should engage more directly with the possibility that these 5 Hz oscillations involve thalamocortical loops. Prior studies have shown that alpha-like rhythms ($\sim 3-5$ Hz) in V1 depend on thalamic activity, with silencing of the visual thalamus abolishing the rhythm. Delving deeper into this mechanism would strengthen mechanistic interpretation.

We thank the reviewer for this suggestion and have revised our discussion accordingly.

Reviewer #3 (Remarks to the Author):

This study presents compelling evidence that systemic administration of serotonin 5-HT_{2A} receptor agonists enhances both spontaneous and visually evoked 5-Hz oscillations in V1 and RSC of awake mice. By leveraging cortex-wide voltage imaging, the authors provide a rare mesoscale perspective on how hallucinogens modulate inter-areal cortical dynamics. The observation of strengthened V1–RSC coupling under psychedelic treatment is particularly novel and potentially impactful for our understanding of the neural basis of perception and posterior cortical oscillatory states. Their technique is really impressive. This represents a potentially important contribution to systems neuroscience. However, several points merit further clarification and discussion.

We thank the reviewer for the kind and supportive feedback.

1. Interpretation of 5-Hz rhythms in the absence of task engagement

The current paradigm relies on passive viewing, with no behavioral task or valence associated with the visual stimuli. Previous studies have reported that ~5 Hz activity in visual cortex is particularly prominent when animals disengage from visual input (Spyropoulos et al., PNAS, 2018; Han et al., eNeuro, 2019). Thus, the observed oscillations could reflect not familiarity per se, but rather an attentional disengagement or “ignoring” of repetitive stimuli. It remains unclear whether the mice were actively attending to the monitor or whether the stimuli were relegated to peripheral vision. The authors should cite these references and explicitly discuss whether passive viewing may have facilitated a state in which oscillations were enhanced by inattention rather than stimulus familiarity.

We thank the reviewer for these suggestions and have revised accordingly.

2. Behavioral phenotypes of systemic serotonin agonists

SC administration of the drug is well documented to induce hyperactivity, stereotypy, and head-twitch responses in rodents. These behavioral phenotypes suggest a broad alteration of brain and bodily states, beyond local cortical modulation. The authors should acknowledge this literature and consider how systemic behavioral effects could interact with, or confound, the observed cortical oscillations. For example, the drugs are assumed to cross the BBB, could the observed effects alternatively reflect peripheral (e.g., muscular or ocular) actions that secondarily altered behavioral or brain state?

We understand that the reviewer’s comment relates to the acute effect of the drug. Of the possible peripheral actions raised by the reviewer, in the context of our current manuscript, we understood that the reviewer is politely hinting to a general state of arousal that may be associated with changes in – for instance – pupil size.

We agree with the reviewer that this is an important factor to consider and thank the reviewer for raising this point (indeed, it is known in the literature that serotonergic psychedelics can induce pupil dilation). We have revised the discussion section, taking into consideration of such possibility. We believe this possibility is limited, as the lack of effect of drugs like DOI

(which we use) on behavioral measures relating to brain state has been previously reported in mice under head-fixed imaging conditions, for instance as Supplementary Figure 2 in Michael et al (reproduced here for the reviewers' convenience).

Figure S2: Baseline and post-drug measures relating to behavioral state. Related to Figure 2. A) Total fraction of experiment time spent running before (pre) and after (post) administration of saline or DOI for each group during two-photon imaging. Open circles connected by dotted lines represent individual animals and closed circles connected by thick lines with error bars are group mean \pm SEM. Significance from paired t-tests are reported above each group plot. B) Average pupil diameter normalized to length of the animal's eye before and after drug administration. Black data represent stationary and red represent running periods. Open circles connected by dotted lines represent individual animals and closed circles connected by thick lines with error bars are group mean \pm SEM.

Figure R3. Supplementary Figure 2 from Michael, Parker & Niell (2017), PMID: 30917304

3. Retinotopic coverage of the stimulus

The visual stimuli used covered $\sim 50-70^\circ$ of visual angle, while the mouse visual field extends beyond 260° . Thus, a substantial portion of the mouse's visual field remained unstimulated. If animals were predominantly processing the stimuli in their peripheral vision, the functional interpretation of the evoked oscillations might differ from what would be observed under full-field or behaviorally salient conditions. Discussion of this limitation, and how it constrains interpretation of "whole V1" dynamics, would strengthen the manuscript.

We appreciate the reviewer's concern. However, we found evoked visual responses across entire V1 as functionally mapped into the Allen Mouse Brain Atlas. This suggests a retinal coverage of at least significant portions of the peripheral visual field. We have amended the discussion to address this.

4. Spatiotemporal patterns of oscillatory propagation

The supplemental videos reveal impressive traveling wave–like propagation of oscillations. A more quantitative analysis of spatiotemporal phase relationships, perhaps using statistical parametric mapping (SPM) across the cortex, would greatly enrich the characterization. This would also clarify whether frontal/motor regions show consistent ~5 Hz phase dynamics relative to V1 and RSC. In short, what is the spatial/spatio-temporal span of 5 Hz rhythm?

We thank the reviewer for raising this important point, which this reviewer shares with reviewer #2.

We have now analyzed the 5-Hz power pixel-wise across several additional defined areas across the cortex, including additional sensory region (barrel cortex) and motor regions (primary and secondary motor cortical regions). Our previous reported observations remain the same: we did not find significant 5-Hz oscillations as strong as that found in V1 and RSC.

This new analysis is provided on page 4 of the current document and in the revised manuscript as new Supplementary Figure 4 c-d.

5. Functional origin of the oscillations

The authors report facilitation of 5-Hz oscillations, in contrast to hippocampal theta where serotonin typically suppresses its power. This discrepancy deserves further discussion. Why should visual oscillations be amplified while hippocampal theta is dampened?

The reviewer raised an interesting point. A recent study by Zimmerman et al showed that hippocampal oscillations emerges independently to that observed in V1. It is therefore possible that the oscillations in these areas are regulated bidirectionally by serotonergic signaling. Another (additive) possibility may be that that preferential activation of serotonin receptor subtype 2A has a different effect to broadly activating across serotonin receptor subtypes.

Unfortunately our imaging approach did not cover the hippocampus and therefore our datasets cannot resolve this question empirically. Future studies (beyond the scope of the current paper) may be designed to experimentally verify this.

If the oscillations are not stimulus-driven, what is their functional role? Recent studies suggest theta-range rhythms may emerge from sensorimotor sampling frequency (Joshi et al., Nature, 2023; Forli et al., Nature, 2025; Bagur et al., Nat Comm, 2021; Karalis & Sirota, Nat Comm, 2022). In visual domain, in humans, inter-saccadic intervals cluster around ~200 ms (~5 Hz; Otero-Millan et al., J Vision, 2008). Could ocular drift or microsaccades induced by the stimulus be contributing to the observed rhythmicity? Clarifying the functional and physiological origin of these oscillations would strengthen the interpretation. I believe the origin of the rhythm is endogenous, but I believe future readers would wonder what is the role and origin of the rhythm.

While recent work showed that mice (which are non-foveal) can make saccade-like gaze shifts, (<https://elifesciences.org/articles/73081>) the underlying mechanisms are different from

the saccades observed in humans (which are foveal). We can only speculate on the function and mechanistic drive of the 5Hz oscillations. Our discussion is now ammended to address these points.

6. Network mechanism of oscillation generation

From a network-model perspective, it would be valuable to elaborate on the putative circuit mechanisms. 5-HT_{2A} activation is known to modulate pyramidal cell dendritic excitability and interneuron interactions, which modulates slow oscillations (de Filippo et al., eLife, 2021). At the same time, recent reports indicate that ~5 Hz rhythms in visual cortex can originate locally within V1/V4 (Zimmerman et al., Curr Biol, 2025; Kienitz et al., Curr Biol, 2021). A discussion integrating these findings—whether the rhythm reflects a pyramidal–interneuron loop, a dendritic resonance effect, PING- or ING-based oscillator circuits, a cascade into RSC, or a rhythmic outside source—would provide important mechanistic context.

We thank the reviewer for the suggestions on how to expand our discussion and integration with previous literature. We added these points to our discussion.